# Impact of institutional support and green knowledge transfer on university students' absorptive capacity and green entrepreneurial behavior: The moderating role of environmental responsibility

**Yejun Yang** [ORCID]*

Office of Academic Affairs, Nantong University, Nantong, Jiangsu, China

\* yejunyang1988@163.com

## Abstract

### Aim/objective

Given an escalated interest in fostering environmental protection, scholars have associated green entrepreneurial behavior as a stimulating factor and the cornerstone of green entrepreneurial performance. Nevertheless, the underlying mechanism that nurtures university students' green entrepreneurial behavior is yet to be explored in the extant literature. Our study proposes the antecedent effects of institutional support and green knowledge transfer to enhance university students' green entrepreneurial behavior. Moreover, we also expand the boundary conditions of these relationships and suggest the mediating effect of university students' absorptive capacity and the moderating effect of environmental responsibility.

### Methodology

The study samples university graduates in Chinese universities ($N = 434$) by adopting a lagged research design spanning over three months. We assessed the proposed model through the multivariate analytical technique.

### Findings

The findings indicate that institutional support and green knowledge transfer significantly elevate university students' green entrepreneurial behavior. Further, these relationships are intervened considerably through absorptive capacity's mediating effect and environmental responsibility's moderating effect.

### Implications

By investigating the crucial roles of institutional support and green knowledge transfer in culminating university students' green entrepreneurial behavior, our study extends the

**Data Availability Statement:** The data obtained from the survey participants is considered confidential. In adherence to our commitment to

maintaining the highest standards of confidentiality, we are unable to share this data. Requests for data access can be sent to: Name: Jing XU Institution: Office of Academic Affairs, Nantong University Address: Nantong 226000, Jiangsu, China Email: xvjing@ntu.edu.cn Phone: +86 13912276103.

**Funding:** The authors received no specific funding for this work.

**Competing interests:** The authors have declared that no competing interests exist.

boundary conditions of these relationships and investigates the hitherto unexplored moderated mediation model.

## Introduction

The critical imperative for environmental preservation has galvanized a wave of scholarly investigation into sustainable solutions. This emerging focus has led to the advent and expansion of green entrepreneurship, a concept that synergizes the conservation of flora and fauna with economic development. Pioneering studies by Li *et al.* [1], Bapoo *et al.* [2], Pascucci *et al.* [3], and Savastano *et al.* [4] highlight the significance of green startups in fostering social, economic, and ecological sustainability. These enterprises not only contribute to economic growth but are instrumental in combating deforestation, enhancing environmental quality, and fortifying ecosystem protection [3].

Green entrepreneurship, an evolving and dynamic field, blends environmental concerns with entrepreneurial endeavors. The terminologies of "environment" and "entrepreneurship" were initially merged in the 1990s [5], and since then, the operational definition and scope of green entrepreneurship have been subjects of scholarly debate and development [6]. Gast *et al.* [7] highlighted the diversity of terms used to describe this phenomenon, including "eco-entrepreneurship," "sustainable entrepreneurship," "environment entrepreneurship," and "ecologically sustainable entrepreneurship." Berle [8] conceptualized green entrepreneurship as the development of products and services adhering to environmental preservation principles, emphasizing an ecology-driven business approach over a purely profit-driven one. Expanding this concept, Schaper [9] integrated elements of creativity, innovation, and market orientation, focusing on ecological protection management and the promotion of cleaner, greener production technologies.

The global context of green entrepreneurship is also evolving rapidly. Nations worldwide recognize the need to balance economic, social, and environmental considerations and are actively promoting green entrepreneurship [10, 11]. In China, for example, there has been a concerted effort to develop policies that regulate greenhouse emissions and promote ecological initiatives, with local government and institutional support playing a pivotal role [6, 12]. A testament to these efforts was the "Internet+" competition in 2017, which engaged over 1.5 million students from 2241 educational institutions, leading to the establishment of numerous green startups [1].

Despite the growing importance of green entrepreneurship, research focusing on university students' green entrepreneurial behavior (GEB) remains limited. The role of students in green startups is crucial, as highlighted by Shabeeb *et al.* [13], yet further investigation is needed to understand the factors motivating GEB among this group [1]. It is posited that internal factors like absorptive capacity (AC) and entrepreneurial responsiveness (ER), along with external factors such as institutional support (IS) and green knowledge transfer (GKT), play significant roles in shaping GEB [14]. Under the ability-motivation-opportunity (AMO) theory, this study proposes that IS and GKT are external factors fostering GEB, AC acts as an internal facilitator, and ER serves as a moderator in this dynamic.

This research aims to explore the extent to which IS and GKT contribute to enhancing university students' GEB. It also seeks to understand how AC mediates the relationship between IS, GKT, and GEB, and the role of ER in moderating these relationships. Such insights are vital for academic institutions and policymakers striving to nurture a new generation of green entrepreneurs.

The study's contribution is multifaceted. It expands the academic discourse on green entrepreneurship by focusing on the underexplored demographic of university students. It integrates theoretical constructs from the AMO theory to examine the antecedents of GEB. Moreover, the research provides practical implications for fostering an educational environment conducive to green entrepreneurial initiatives.

Hence, the following research questions will thus be addressed by the current investigation:

1. To what extent (b) IS and (b) GKT are helpful in elevating university students' GEB?

2. To what extent AC mediates the links between (b) IS and GEB and (b) GKT and GEB?

3. To what extent ER responsiveness intervenes in the association between IS and GKT on university students' GEB via AC?

The remaining study is structured to find the answers to the aforementioned research questions as follows: the subsequent section discusses theoretical underpinning and explains the relationship among variables; participants and measurement scales are discussed in the methodology section, which follows results interpretation, discussion, and implications of the study.

## Theoretical background and hypotheses

### Green entrepreneurship

Green entrepreneurship, as defined by Schaltegger and Wagner [15], is the process of identifying, developing, and exploiting business opportunities that are an impetus for a sustainable future. It encapsulates not only profit-making ventures but also values environmental protection and social responsibility. O'Neill and Gibbs [16] emphasize how green entrepreneurship is critical in driving the transition to a sustainable economy, by innovating eco-friendly technologies and sustainable business practices. The role of green entrepreneurs in promoting environmental sustainability is also highlighted by Savastano *et al.* [4], who argue that these entrepreneurs are key players in addressing ecological challenges through market-based solutions. Furthermore, Shabeeb *et al.* [13] discuss how market failures related to environmental goods create opportunities for green entrepreneurs to intervene and create value. This body of work collectively emphasizes the growing importance of green entrepreneurship in the contemporary business landscape, especially in the context of increasing environmental concerns and the global push towards sustainability. Therefore, enriching the study with these perspectives can provide a deeper understanding of the implications of fostering green entrepreneurial behavior, particularly among university students who are at the forefront of shaping future business practices.

### Ability-motivation-opportunity theory

Relying on the AMO theory, we predict the antecedent effects of contextual factors: IS and GKT on university students' GEB through AC's mediating effect and ER's moderating effect. The AMO theory [17], purports that individual behaviors can be shaped through maneuvering several factors such as ability, motivation, and opportunity. Ability represents a person's experience, skills, knowledge, and other capabilities, *e.g.*, stamina, endurance, health, etc., necessary to execute specific behavior [17]. Motivation involves the degree of willingness to engage in the particular behavior [17]. While, opportunity includes factors assisting the smooth execution of that behavior [17]. The use of AMO framework in the entrepreneurship research has been widely recognized. For instance, Mia *et al.* [18] applied the AMO model to predict the impact of skills, incentives, and entrepreneurial education in facilitating green

entrepreneurship and social change. Likewise, Raza *et al.* [19] employed the AMO framework to theorize the impact of ability, motivation, and opportunity on university students' entrepreneurial intentions.

As per the AMO framework, factors that contribute to the development of ability, motivation, and opportunity among individuals are usually specified with respect to specific tasks [17]. The current study relies on the AMO model to explain the link between university students' individual and contextual factors and GEB. In this milieu, students perform two specific tasks. First task involves the knowledge acquisition in terms of transformation of knowledge from explicit to tacit, which the educational institutions instill in their students by the means of GKT. In addition to that IS plays a crucial role in fostering abilities, knowledge, and skills for carrying out green initiatives. Further, because students enrolled in the business programs are trained and developed in such as way, they are more likely to become future entrepreneurs, they develop a degree of motivation, *i.e.*, self-driven motivation to transform the acquired knowledge into practical execution in terms of entrepreneurial endeavors. In this regard, ER functions as an inner drive to get engaged in behaviors that may facilitate the translation of students' green initiatives into realism. Further, AC infuses students to acquire, assimilate, and exploit knowledge from universities to their actual business.

## Hypotheses

### Linking institutional support with absorptive capacity

Local institutional environment plays an imperative role in influencing the behavior of entrepreneurs [20]. This is because entrepreneurs need to strive to line up their behavior with central government requirements, values, and social rules to attain IS [21]. IS refers to as "a variety of different types of policy support that government departments provide for entrepreneurs and enterprises" [6]. Such support may include licenses, market information, tax rebates, development subsidies, research, and other related resources that will aid in addressing green entrepreneurs resource deficiencies. Thus, entrepreneurs pursuing green initiatives may be able to obtain key resources through high level of IS [20]. For instance, a study conducted by Meek *et al.* [22] on solar energy enterprises in the US found that the creation of new enterprises in the industry are largely affected by public consumption and government investments. Similarly, another host of researchers employed the resource theory to explain Chinese university students' entrepreneurship phenomenon. Their study stressed the significant role of institutional environment in facilitating entrepreneurial activities among university students. China, as an eminent reflection of the transitional economy, has a substantial bondage between entrepreneurship and government [6]. Examples of such support include large amount of information and knowledge resources through institutional support, e.g., "Innovation and Entrepreneurship of National Strategy in China" operational since 2016; and "Building Beautiful China" effective from 2012 [6]. Moreover, other institutional support includes, e.g., preferential purchase of green products, facilitating green enterprise registration, and alleviated green entry procedures [23].

We speculate that the impacts of institutional support on GEB is casted through university students' AC. AC is defined as "the ability of individuals to acquire, assimilate, transform, and exploit knowledge or obtain information" [24]. AC has primarily been studied by the Entrepreneurship and Business scholars at organizational and groups' levels [25]. Albeit, group or organization's AC is a composite of individual skills and knowledge [24]. According to Bouguerra *et al.* [26], personal knowledge, skills, and observations serve as the chief fundamentals of firm level performance. Since the central premise of this research revolves around individuals, who acquire, assimilate, and transform particular skills, knowledge, and abilities, that are

required to pursue their entrepreneurial endeavors [24], the current study encapsulates university students' AC as the key mechanism of to explain and manage entrepreneurial behavior. Hence, in our study's context, AC refers to "the ability of students to exploit knowledge" during their university education, from their schools and other relevant sources, i.e., institutional support, "to recognize its value, assimilate it, combine it with existing knowledge, and apply it [in pursuing entrepreneurial careers]" [27]. Thus, relying on the AMO theory, we view students' AC in terms of knowledge-based capacity [28]. Through this ability, university students can transform and develop this acquired knowledge through institutional support and cultivate an enhanced entrepreneurial mindset [29]. The degree of knowledge acquired from the institutional support and the degree of knowledge applied in the execution of green entrepreneurship depends on the level of university students' AC. Therefore,

H1a. Institutional support has a positive effect on absorptive capacity.

## Linking green knowledge transfer with absorptive capacity

We also envisage the antecedent effect of GKT on university students' AC. A massive stream of research emphasizes the role of university knowledge as an imperative source of organizational innovation and scholars have devoted their attention to examine the key features of this knowledge transition from university to industry [24], particularly entrepreneurship [30]. In this regard, literature has identified three distinct factors as the building blocks of this knowledge transfer: university features, industry features, and transfer channel features [31]. In terms of the transfer channels, researchers have widely recognized and explored a number of channels that facilitate this transition, e.g., personal exchange, patents, publications, consulting services, licensing, etc. [32]. However, university students acting as stimulators of knowledge transfer from university to industry has rarely been investigated [30]. We speculate that university students' GKT from university to industry, i.e., commencing a green enterprise, depends on university students' AC that facilitates the smooth culmination of the green knowledge that students acquire from their universities to their green ventures.

Balle *et al.* [33] contended that knowledge transfer seeks to find a medium between a sender and a recipient, wherein, the sender provides knowledge and the recipient acquires it and utilize this transferred knowledge. Hence, the characteristics of knowledge transfer not only depend on the source but also put equal emphasis on the recipient to absorb and reproduce this knowledge whenever it is necessary [31]. In this perspective, students' AC functions as a channel between the sender, i.e., university that provides green knowledge to their students, and the recipient, i.e., students who acquire this knowledge and assimilate it and make it readily available for reuse through AC. The AMO theory supports this theoretical deduction that universities transfer green knowledge in their students, thereby enhancing their knowledge, skills, and abilities, which nurture their inner drive to identify and exploit opportunities in terms of green enterprise development. Therefore,

H1b. Green knowledge transfer has a positive effect on absorptive capacity.

## Linking absorptive capacity with green entrepreneurial behavior

Subsequently, the knowledge acquired by the university students through institutional and university support, assists in the transformation of GEB. It is argued that entrepreneurship and innovation are the upshots of efficient knowledge transfer [30]. Hence, both learning and innovation are based on AC [29]. Chung *et al.* [25] identified AC as the key source of entrepreneurial capabilities that possess the abilities to absorb internal and external knowledge.

According to Chao and Yu [24], AC constitutes a set of skills that revolve around individuals' cognitive, digestive, and application capabilities. This is further endorsed by Bouguerra *et al.* [26] that there exists a direct relationship between AC and innovation and creativity. Similarly, Akbari [28] sanctioned that AC results in boosting innovation and knowledge creation, which are the keystones of entrepreneurship [25]. Moreover, Miller *et al.* [32] have associated AC with firms' dynamic capabilities that leverage enterprises to sustain in turbulent times. We hypothesize that for green entrepreneurs to prepare and conduct business in a dynamic and challenging environment, knowledge is the key to successful execution of entrepreneurship and entrepreneurial abilities. Hence, AC provides a source of competitive advantage to young entrepreneurs to fuel GEB through continuous acquisition, assimilation, transformation, and utilization of new knowledge adequately. In a recent study conducted by Chung *et al.* [25], the authors found significant role of AC in predicting entrepreneurial intentions. There is a wide agreement that substantiates that intentions are a strong predictor of behavior [6, 11, 13, 19]. Therefore,

H2. Absorptive capacity has a positive effect on green entrepreneurial behavior.

## The mediating effect of absorptive capacity

Combining the hypotheses 1a & b and 2, we deduce the mediating effect of AC between IS and GEB and GKT and GEB. The AMO theory supports this assumption that university students acquire abundant knowledge, skills, and abilities, that may be helpful in identification and exploitation of opportunities by means of commencing a green enterprise. Hence, AC plays a pivotal role in facilitating the acquisition, assimilation, transformation, and culmination of the valuable resource: knowledge into execution, i.e., GEB. Therefore,

H3. Absorptive capacity mediates the relationships between (a) institutional support and green entrepreneurial behavior, and (b) green knowledge transfer and green entrepreneurial behavior.

## Moderating effect of environmental responsiveness

Research indicates that a sense of responsibility towards environment plays an essential role in enhanced environmental familiarization through knowledge acquisition [24]. ER that triggers individuals' sense of ecological responsiveness represents such a context Li *et al.* [34]. According to Mansoor and Wijaksana [35], ER has been highlighted as a contextual moderator in the literature of pro-environmental behaviors. ER is defined as "individuals' determination, spirit of dedication, and sense of responsibility to prevent environmental degradation and solve ecological and environmental problems" [24, p. 4]. Further, ER reflects the bearing of individuals' obligations and responsibilities in coping with environmental issues through managing, protecting, utilizing, and exploiting resources [36]. In addition, individuals who draw on their ER characteristics are internally driven and they are prompted to assume ER and manifest pro-environmental behaviors [37]. This is validated by Chao and Yu [24] that ER engenders individuals' determination to mitigate environmental and ecological problems. Much research in the past has conceptualized ER from organizational perspective, however, individual's ER stance has rarely been tested earlier. The valuable exception of Chao and Yu's [24] research highlight the need for further inquiry on ER at individual level. We draw on the AMO theory and project that ER intervenes individuals' green knowledge acquisition from IS and university support: GKT. We suggest that GKT occurs on an exchange relationship between a source and a recipient. In our study's perspective, universities and institutions extend such support by

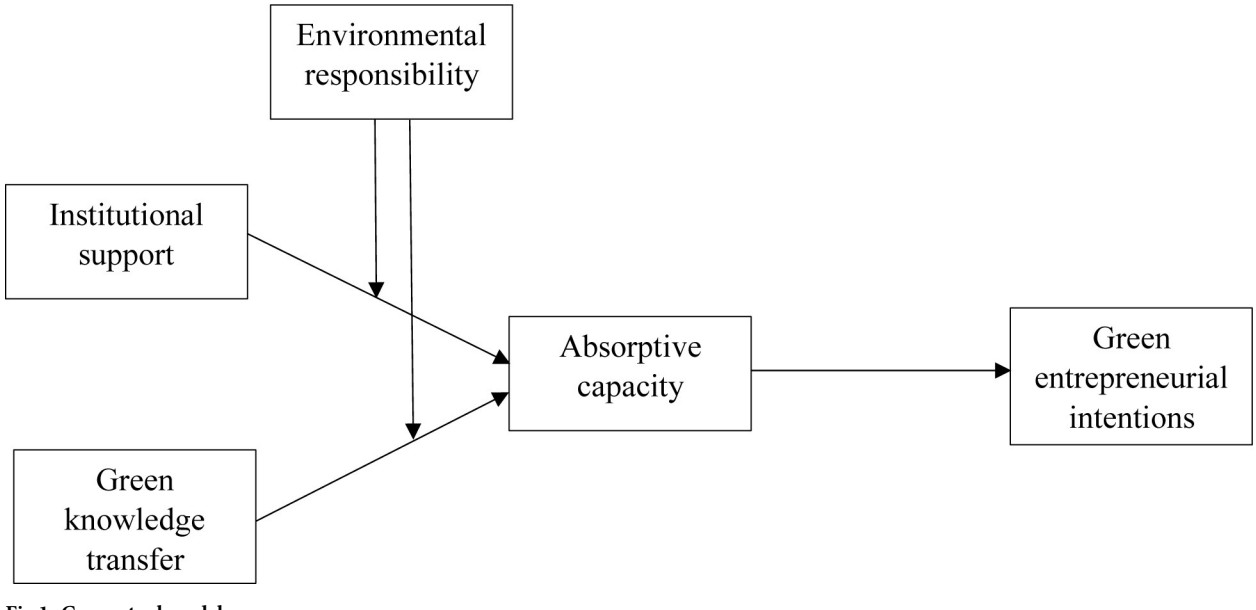

**Fig 1. Conceptual model.**

instilling and awakening a sense of environmentalism among students. When students face a dilemma between personal and social interests, they embark upon their ER attitude to demonstrate and implement environmental protection behaviors [24]. This, in turn, improves the cultivation and an efficient transition of green knowledge, in the form of AC, prompted by IS and GKT. This is further supported by Pham and Khanh [38], who argued that ER enhanced individuals' determination, spirit, and courage to address environmental issues. Therefore, students with high levels of ER will more likely be transforming IS and GKT in high levels of AC and vice versa. Hence,

H4. The relationships between (a) institutional support and absorptive capacity and (b) green knowledge transfer and absorptive capacity, will be moderated by environmental responsiveness. The higher the environmental responsiveness, the stronger the relations.

Taken together, ER moderating the direct relationship between IS and GKT and university students' AC, and AC mediating the relationship between IS and GKT and GEB, we also propose the moderating effect of ER in these relations. We hypothesize that students with high ER are more likely to transform IS and GKT into enhanced GEB through AC than their counterparts (shown in Fig 1). Therefore,

H5. The indirect relationships between (a) institutional support and green entrepreneurial behavior through absorptive capacity and (b) green knowledge transfer and green entrepreneurial behavior through absorptive capacity, will be moderated by environmental responsiveness. The higher the environmental responsiveness, the stronger the relations.

## Methodology

### Participants and procedure

A sample of randomly selected university graduates in the Chinese universities has participated in this study by employing a lagged research design with an interval period of 1 month between each wave. The targeted university graduates have their major in entrepreneurship,

business, ecommerce, environmental engineering, and economic management. The rationale for the selection of university graduates is based on the increasing inclination of university students to respond to rising environmental issues through ecopreneurship as a result of national "Strategy of Building a Beautiful China" [6]. Further, researchers and practitioners have serious concerns to examine university students' entrepreneurial behavior because prior research has shown a significant gap between students' desires to pursue entrepreneurial careers and their actual engagement in startups [13, 39]. We approached the university graduates with the assistance of researchers and managerial personnel at the concerned universities. The respondents were given questionnaires and cover letters through which we collected their responses and informed about the study's purpose and procedure to participate in the survey and generate the keys for their responses and identification. Data collection spanned over 3 months from November 2022 to January 2023, and participants provided responses for IS and GKT and ER in the first wave; AC in the second wave, and GEB in the third wave. We collected 416 valid responses out of 500 distributed questionnaires to the study's participants in multiple waves. Out of which, 62% were male and 38% were female participants with an average age of 31 (with a mean SD: 4.56). 44%, 36%, and 20% of the respondents were in the services, manufacturing, and hospitality businesses.

## Measurement scales

We examined the relationship between IS and GKT as antecedents of students' GEB through AC's mediating effect and ER's moderating effect. Guided by the study's purpose, a quantitative approach is employed in this study, and data for empirical analysis of the structural paths are collected using questionnaires. The questionnaire contains two parts: part I gathers information about the demographic profile of the respondents; part II collects their responses about the study's variables. To collect participants' responses about IS, GKT, AC, ER, and GEB, we adopted previous measurement scales and modified them according to the study's context. Turker and Selçuk's [40] scale of 4 items was adapted to examine IS. The sample items are "In China, entrepreneurs are encouraged and supported by public and non-governmental institutions (*e.g.*, district and municipal assemblies, trade associations)". Ko *et al.'s* [41] scale of 3 items was adapted to examine GKT. The sample items are "I have acquired a lot of green knowledge and skills applicable to my current entrepreneurial behavior" and "I have acquired a lot of green knowledge and skills that helps me to enhance my entrepreneurial behavior". Tho's [31] scale of 4 items, was adapted to examine AC. The sample items are "I have the ability to absorb the new knowledge and skills provided by my instructors" and "I have the ability to integrate the new knowledge and skills provided by my instructors with my prior knowledge". Yang *et al.'s* [42] scale of 4 items was adapted to examine ER. The sample items are "I am willing to increase my purchase of eco-labeled goods to help solve environmental problems" and "I have a responsibility to help solve environmental problems that society faces". Kautonen *et al.'s* [43] scale of 4 items was adapted to examine GEB. The sample item is "started green product/service development".

## Analytical strategy

The study proposed a complex moderated mediation model consisting of two independent variables: IS and GKT, one mediating variable: AC, one dependent variable: GEB, and one moderating variable: ER. To investigate the association among these variables, we employed SmartPLS SEM for examining the outer and inner models. This is in line with the recommendation of Hair *et al.* [44] to validate the outer model prior to inner model estimation. The subsequent section presents the results of the study.

## Results

### Outer model assessment

We hypothesized a "reflective framework" in this study. To measure the "reflective outer/measurement model", the study determines "internal consistency" and "convergent and discriminant validity". To measure the "internal consistency", the study assesses the "composite reliability (CR)" and "Cronbach's alpha" metrics. Nunnally and Bernstein [45] provided the minimum acceptable value for CR and Cronbach's alpha to be 0.70. Table 1 illustrates that IS (CR = 0.912, α = 0.868), GKT (CR = 0.880, α = 0.845), AC (CR = 0.876, α = 0.834), ER (CR = 0.895, α = 0.854), and GEB (CR = 0.903, α = 0.872) have values greater than the minimum acceptable thresholds. Furthermore, to measure the "convergent validity", the study determines the "average variance extracted (AVE)" and "outer loadings", considering the minimum threshold value of 0.50. Table 1 shows that all the AVE values for IS (0.650), GKT (0.628), AC (0.623), ER (0.643), and GEB (0.678) are above 0.50, thus ensuring convergent validity in the study. Further, the outer loading values support this analysis as the minimum and maximum values for IS (IS2 = 0.754; IS1 = 0.842), GKT (GKT1 = 0.795; GKT2 = 0.802), AC (AC4 = 0.723; AC3 = 0.833), ER (ER2 = 0.764; ER1 = 0.834), and GEB (GEB3 = 0.795; GEB4 = 0.843) are greater than 0.50.

In addition, the study also measures the discriminant validity to ensure that the inter-construct correlations should not exceed the intra-construct correlations. The study assessed the "discriminant validity" using the "heterotrait-monotrait (HTMT)" criteria under the recommendations of Hair *et al.* [44]; Henseler *et al.* [46]. In order to determine the HTMT ratio, the

**Table 1. Construct validity and reliability.**

| Code | FL | AVE | rho_A | Cronbach's alpha |
|---|---|---|---|---|
| Institutional support | | 0.650 | 0.912 | 0.868 |
| IS1 | 0.842 | | | |
| IS2 | 0.754 | | | |
| IS3 | 0.821 | | | |
| IS4 | 0.805 | | | |
| Green knowledge transfer | | 0.628 | 0.880 | 0.845 |
| GKT1 | 0.795 | | | |
| GKT2 | 0.802 | | | |
| GKT3 | 0.782 | | | |
| Absorptive capacity | | 0.623 | 0.876 | 0.834 |
| AC1 | 0.814 | | | |
| AC2 | 0.783 | | | |
| AC3 | 0.833 | | | |
| AC4 | 0.723 | | | |
| Environmental responsiveness | | 0.643 | 0.895 | 0.854 |
| ER1 | 0.834 | | | |
| ER2 | 0.764 | | | |
| ER3 | 0.821 | | | |
| ER4 | 0.764 | | | |
| Green entrepreneurial behavior | | 0.678 | 0.903 | 0.872 |
| GEB1 | 0.834 | | | |
| GEB2 | 0.822 | | | |
| GEB3 | 0.795 | | | |
| GEB4 | 0.843 | | | |

**Table 2. Heterotrait-monotrait (HTMT) ratio.**

| | IS | GKT | AC | ER | GEB |
|---|---|---|---|---|---|
| IS | | | | | |
| GKT | 0.604 CI.₋₀.₉₀₀ [0.543;0.670] | | | | |
| AC | 0.632 CI.₋₀.₉₀₀ [0.549;0.702] | 0.645 CI.₋₀.₉₀₀ [0.577;0.701] | | | |
| ER | 0.717 CI.₋₀.₉₀₀ [0.652;0.769] | 0.558 CI.₋₀.₉₀₀ [0.459;0.638] | 0.823 CI.₋₀.₉₀₀ [0.757;0.881] | | |
| GEB | 0.700 CI.₋₀.₉₀₀ [0.622;0.780] | 0.747 CI.₋₀.₉₀₀ [0.680;0.794] | 0.456 CI.₋₀.₉₀₀ [0.482;0.619] | 0.601 CI.₋₀.₉₀₀ [0.532;0.679] | |

study processed the "bias-corrected and accelerated (BCa)" bootstrapping technique with a resample of 5,000, using a one-tailed $t$-test at a 90% significance level. This will help the researchers to yield to estimates with an error probability of 5% using the two-tailed. Henseler *et al*. [47] recommended the maximum threshold value of HTMT.₈₅. Results shown in Table 2 reveal that all the HTMT values are lesser the acceptable limit of HTMT.₈₅, thereby validating discriminant validity.

## Inner model assessment

After assessing the measurement model, the study determines the structural model using a BCa bootstrapping technique on a resample of 5,000 in order to generate the $t > 1.96$) and $p < 0.05$ values for measuring the path coefficients ($\beta$). Moreover, the study also measures the "coefficient of determination ($R^2$)", "predictive relevance ($Q^2$)", and "effect size ($f^2$)", for estimating the relationship among the latent variables. Hair *et al*. [44] suggested that the researchers should also determine the effect size ($f^2$) in addition to estimating the $R^2$ value, which reflects "the change in the $R^2$ value when a specified exogenous construct is omitted from the model can be used to evaluate whether the omitted construct has a substantive impact on the endogenous constructs" (p. 211). According to Cohen [48], $f^2$ values of 0.02, 0.15, and 0.35 represent "small", "medium", and "large" effects. *Results* presented in Table 3 show that IS has a significant positive association with AC ($\beta = 0.482$; $t = 8.243$; $p = 0.000$; $f^2 = 0.252$), with a medium effect size; and GKT has a significant positive association with AC ($\beta = 0.554$; $t = 9.259$; $p = 0.000$; $f^2 = 0.232$), with a medium size. The analysis supports Hypotheses 1a and b.

**Table 3. Effects on endogenous variables.**

| Hypotheses | $\beta$ | CI (5%, 95%) | SE | $t$-value | $p$-value | $f^2$ |
|---|---|---|---|---|---|---|
| H1a IS -> AC | 0.482*** | (0.420,0.563) | 0.062 | 8.243 | 0.000 | 0.252 |
| H1b GKT -> AC | 0.554*** | (0.476,0.621) | 0.048 | 9.259 | 0.000 | 0.232 |
| H2 AC -> GEB | 0.489*** | (0.419,0.560) | 0.051 | 10.253 | 0.000 | 0.284 |
| H3a IS -> AC -> GEB | 0.424*** | (0.365,0.482) | 0.078 | 7.728 | 0.000 | 0.352 |
| H3b GKT -> AC -> GEB | 0.492*** | (0.423,0.577) | 0.050 | 4.493 | 0.001 | 0.282 |
| H4a ER_IS -> AC | 0.330*** | (0.276,0.390) | 0.064 | 6.822 | 0.000 | 0.258 |
| H4b ER_GKT -> AC | 0.289*** | (0.221,0.367) | 0.072 | 8.478 | 0.000 | 0.231 |
| H5a ER_IS -> AC -> GEB | 0.367*** | (0.319,0.431) | 0.080 | 11.489 | 0.000 | 0.381 |
| H5b ER_GKT -> AC -> GEB | 0.410*** | (0.346,0.478) | 0.042 | 3.490 | 0.003 | 0.212 |

**Table 4. $R^2$ and $Q^2$ values for latent variables.**

| Endogenous latent variables | $R^2$ values | $Q^2$ |
|---|---|---|
| AC | 0.550 | 0.332 |
| GEB | 0.484 | 0.421 |

In addition, the study predicted that AC mediates the direct effect of IS on GEB and GKT and GEB. The study assesses the mediation analysis using Zhao *et al.'s* [49] mediation approach. In order to yield the *t* and *p* values, the study ran BCa bootstrapping of 5,000 resamples. Results illustrated in Table 3 show that the indirect effects of IS on GEB through AC and GKT on GEB through AC are significant with CIs (0.365, 0.482) and (0.423, 0.577) respectively. As both the direct and indirect effects are significant, the analysis reveals the complementary mediating role of AC between IS and GEB and GKT and GEB.

Furthermore, to measure the moderation analysis, the study assesses the "two-stage" approach for measuring the interaction effect of ER and IS and ER and GKT on AC. The study measures the effect size using BCa bootstrapping of 5,000 resamples. Results produced in Table 3 show that the interaction term (ER_IS) has a significant direct effect on AC based on *t* > 1.96 and *p* < 0.05 values ($\beta$ = 0.330; *t* = 6.882; *p* = 0.000; $f^2$ = 0.258); and a significant indirect effect on GEB through AC based on *t* > 1.96 and *p* < 0.05 values ($\beta$ = 0.367; *t* = 11.489; *p* = 0.000; $f^2$ = 0.381). The results show the medium and large effect sizes of the impact of the interaction term (ER_IS) on AC and GEB via AC. Similarly, the interaction term (ER_GKT) has a significant effect on AC based on *t* > 1.96 and *p* < 0.05 values ($\beta$ = 0.289; *t* = 8.478; *p* = 0.000; $f^2$ = 0.231); and a significant indirect effect on GEB through AC based on *t* > 1.96 and *p* < 0.05 values ($\beta$ = 0.410; *t* = 3.490; *p* = 0.000; $f^2$ = 0.212). The results show the medium effect sizes of the impact of the interaction term (ER_GKT) on AC and GEB via AC. The analysis supports Hypothesis 4 & 5a & b.

In addition, the values of $R^2$ and $Q^2$ reported in Table 4 show the variance explained and predictive capability of the model. The $R^2$ values of 0.550 for AC and 0.484 for GEB show the moderate impact. Further, $Q^2$ the validate the predictive capability of the model as the values are greater than 0.

Furthermore, the study assesses the graphical representation of the interaction effect using a "2-way unstandardized" approach to measure the interaction effect of ER and IS and ER and GKT on GEB through AC. Results presented in Figs 2–5 show that at high levels of ER, the associations between IS and GEB via AC; and GKT and GEB via AC are stronger. On the other hand, at low levels of ER, the IS–GEB link via AC and GKT–GEB link via AC, are weaker.

## Discussion

The current study assessed the relationship between university students' external factors: IS and GKT and GEB. Further, the study envisaged the mediating effect of AC and the moderating effect of ER, serving as the internal factors, between IS and GEB and GKT and GEB. As projected, IS and GKT positively associated with GEB. The results further revealed that AC significantly mediated the links between IS and GEB and GKT and GEB. Further, ER significantly moderated these relations such that these links are stronger at high levels of ER than at low levels. The study's findings are in harmony with previous literature. For instance:

The confirmation of Hypotheses 1a and 1b reinforces the critical role of IS and GKT in enhancing AC. This aligns with the findings of previous studies which have indicated that both IS and GKT are pivotal in developing the ability of individuals and organizations to

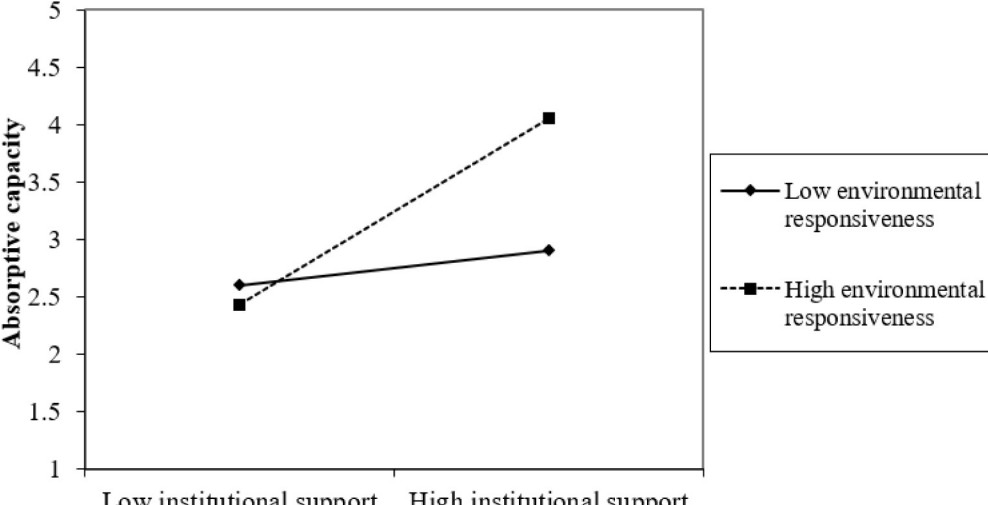

**Fig 2. Moderating effect of institutional support on absorptive capacity.**

recognize, assimilate, and utilize new knowledge [50, 51]. In the context of green entrepreneurship, this suggests that support from institutions and the transfer of green knowledge are essential in equipping students with the necessary capabilities to absorb and apply environmental information effectively.

The positive effect of AC on GEB, as suggested in Hypothesis 2, is corroborated by the research findings. This echoes the assertions of prior research that emphasizes the role of AC in entrepreneurial activities [25, 52]. The ability to assimilate and apply new knowledge effectively is a crucial determinant of entrepreneurial behavior, particularly in the green sector where innovation and adaptability are key.

The mediation role of AC between IS and GKT on one hand, and GEB on the other, as proposed in Hypothesis 3, is validated by the study. This supports the theoretical framework that

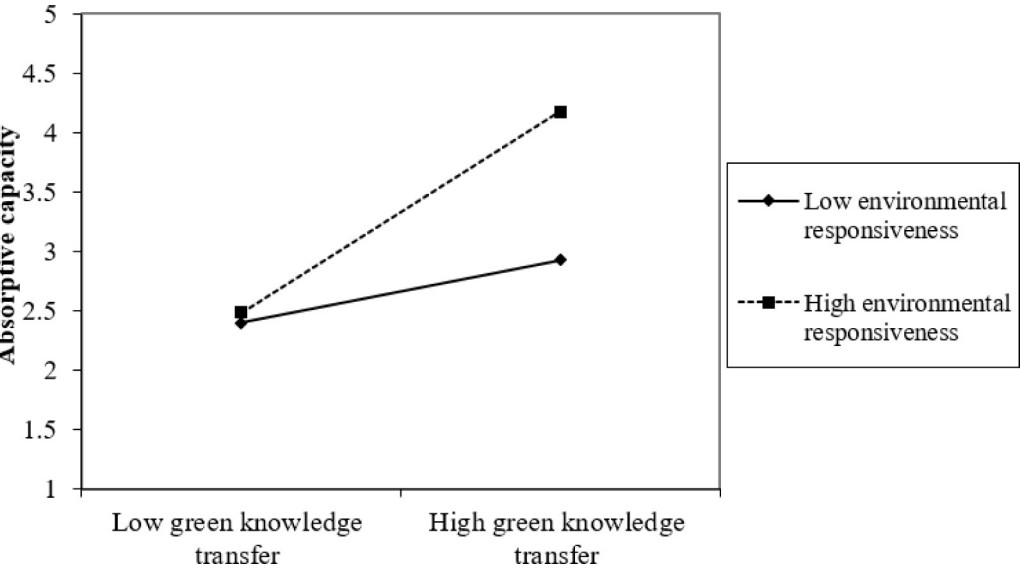

**Fig 3. Moderating effect of green knowledge transfer on absorptive capacity.**

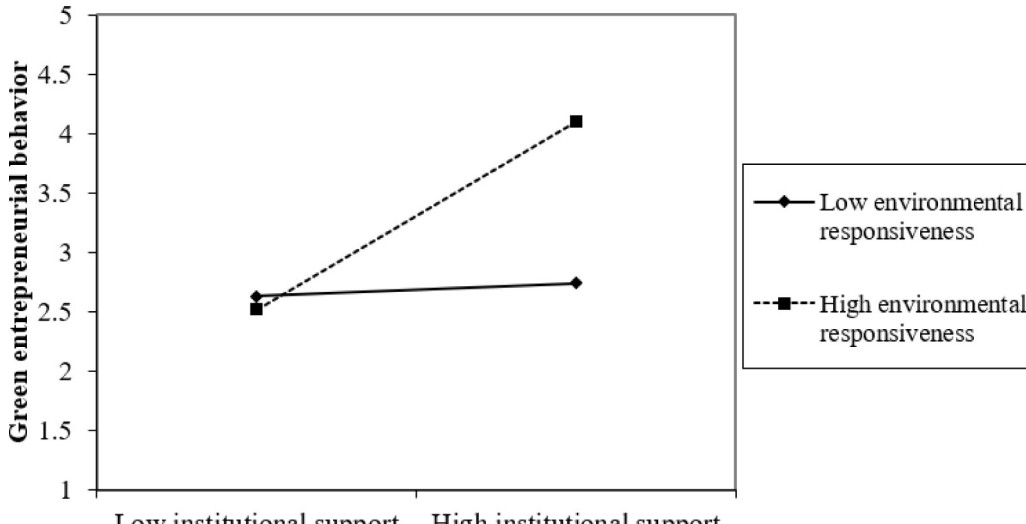

**Fig 4. Moderating effect of institutional support on green entrepreneurial behavior through absorptive capacity.**

posits AC as a crucial link between external resources and entrepreneurial behavior [53]. In the context of green entrepreneurship, this implies that the benefits of institutional support and knowledge transfer are channeled through enhanced AC, leading to more effective entrepreneurial actions.

The moderating role of ER in the relationships between IS and GKT with AC, as hypothesized in H4, is also affirmed. This finding is in line with previous studies that highlight the importance of contextual factors in influencing the impact of external resources on AC [54]. The higher the level of ER, the more pronounced is the effect of IS and GKT on AC, suggesting that a conducive environmental context amplifies the impact of these external factors.

Lastly, the research confirms the moderating role of ER on the indirect relationships between IS and GKT with GEB through AC (H5). This finding aligns with the notion that the

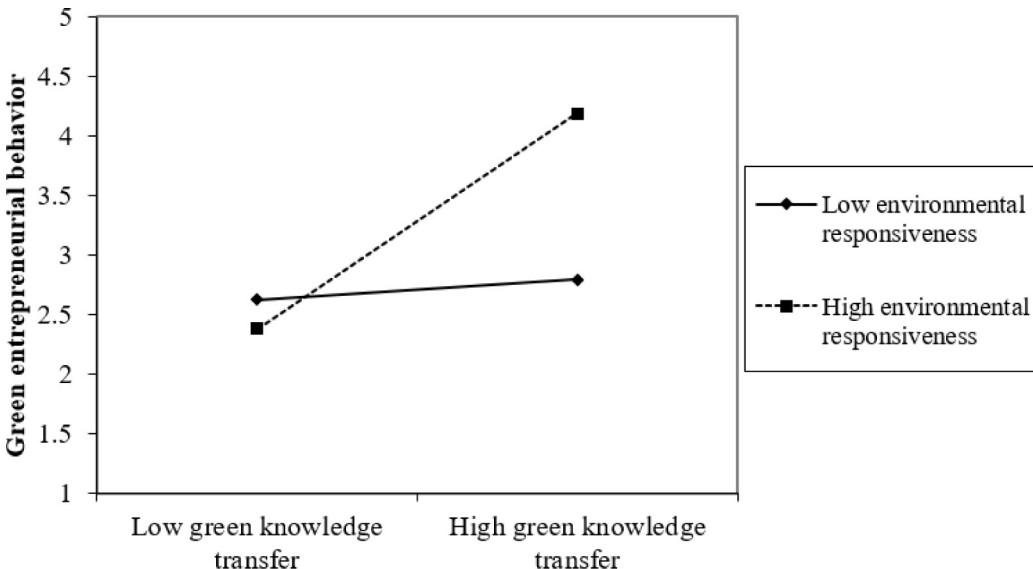

**Fig 5. Moderating effect of green knowledge transfer on green entrepreneurial behavior through absorptive capacity.**

context within which learning and adaptation occur is critical in determining the effectiveness of these processes [31]. In environments with high ER, the pathways from IS and GKT through AC to GEB are strengthened, underscoring the importance of a responsive environmental context in fostering green entrepreneurial initiatives.

## Implications for theory

Our study has several noteworthy implications for theory by bridging the knowledge gaps between IS and GKT and GEB. First, past research in the domain of university students' entrepreneurial behavior has made great strides [39, 40, 55, 56]. However, there has been so far little research on topic applied to green entrepreneurship [1, 13]. Our study highlights the crucial role of external factors such as IS and GKT in fueling university students' GEB. A review of the preliminary inquiry on the antecedents of GEB indicates that majority of previous studies have linked green entrepreneurial intentions as the key facilitator of students' GEB [13, 56, 57]. Further, studies have associated institutional environment and university support as important stimulators of GEB [23]. Our study makes novel contributions by linking IS and GKT as the eminent precursors of GEB. Our findings provide empirical evidence on the link between IS and GKT and GEB, which has never been investigated earlier except by [6], who identified IS as a causal factor between green entrepreneurial intentions and GEB. Moreover, knowledge transfer has lately been identified as a relevant factor and thus only few studies have examined its implications in the industrial context. For instance, Tho [31] examines in-service students' business to industry knowledge transfer and its impact on the firm performance. Thus, this is the first study to assess the relationship between IS and GKT and GEB.

Second, we extended Naushad et al. [21]; Yi [6] studies by investigating the intervening role of AC in the relationship between IS and GKT and GEB. We propose AC as a causal mechanism that explains how do IS and university support: GKT, translate into GEB, which has generally been left unexplored. Relying on the AMO theory, we suggest that IS and GKT can enhance university students' AC, which in turn will foster GEB. In general, our results provide the potential implications of IS and GKT and that their effects on GEB are exerted through AC. Our study contributes to the limited literature on the individual AC and its implications for university students in translating the impact of IS and GKT in enhanced entrepreneurial endeavors. In addition, we further expect the implications of AC spilling over into entrepreneurial innovative and creative performance. There is ample evidence, which suggests that an effective management and application of knowledge is crucial for individuals' creative and innovative behaviors [19, 31]. This merits future studies to explore the mediating role of AC between IS and GKT and entrepreneurs' innovation performance.

Third, our study revealed that the direct association between IS and GKT and AC and the indirect association between IS ang GKT and GEB through AC can be stimulated through an external factor: ER. To the best of our knowledge, no prior research has explored the moderating role of ER in nurturing students' entrepreneurial endeavors. Drawing on the AMO theory, our study found that high levels of ER leverage intrinsic motivation and self-determination among students, thereby strengthening the bond between IS and GKT and AC. Subsequently, the effects of IS and GKT on GEB through AC are more potent at high ER levels than at low ER levels. Thus, another contribution of this study is to examine the boundary effects of ER coupled with IS and GKT on AC and GEB. Specifically, our study not only theoretically postulated the interaction effect of ER and IS and GKT on AC and GEB but also empirically examined the moderating effect of ER in these relationships. In addition to this our study extend the theoretical implications of ER by studying its impact at individual level. However, in response to sustainable development goals, extant scholars have tested the ER construct from

organizational perspective. Our study thus flips the perspective and explores a hitherto under-explored role of ER in leveraging university students to nurture GEB.

Last but not the least, prior research has examined the antecedents of entrepreneurial behaviors from divergent theoretical perspectives, including theory of planned behavior [58], expectancy theory [59], theory of reasoned action [60], resource-based view [61], and institutional theory [62]. However, examining the boundary conditions of GEB from the AMO framework provides a unique and novel contributions to the limited research on the employment of AMO as the theoretical foundation in entrepreneurship research.

## Implications for practice

Our study has numerous important implications for practitioners and policymakers. First, we encourage local and government institutions to extend high institutional support, engage in concurrent facilitating programs, e.g., business incubation centers to the young entrepreneurs, specifically those who are committed to make breakthrough innovations in their products and services offering that may result in enhanced environmentalism. There should be organized awareness programs and training sessions that may result in increased students' inclination toward ecopreneurship. Further, these ventures should be promoted through tax reliefs and lenient startups procedures. Similarly, the role of universities in contributing to student progress and development is equally important. Entrepreneurship is regarded as one of the finest substitutions of professional careers. That is to say, through entrepreneurial initiatives, students can not only mitigate the rising issues of unemployment but also be able to contribute to national economy and progress. Universities should realize the importance of green entrepreneurship and devise strategies, such as designing curricula that empowers students' environmental concerns and knowledge. Universities are an integral part of individuals' knowledge, skills, and abilities accumulation [30]. Students should be given opportunities to learn and hone skills and abilities that will be required to commence their businesses. In this regard, universities should organize trainings, seminars, conferences, and different industry-academia linkage programs that may expose students to latest trends and exposure of the practical worlds. Further, there should be internal business incubation centers that should not only welcome students' infant business ideas but also encourage and promote these ideas through professional counseling, assistance, and guidelines. In addition, a sustainability orientated culture should be nurtured in universities. Students should be given opportunities to voluntary participate in charitable, social, and other environmental activities so as to enhance their dedication and determination to preserve the environment. Last but not the least, our study found the intervening role of AC as a causal mechanism that strengthens these links. We suggest that universities should have a knowledge driven culture that facilitates the transformation of explicit knowledge into tacit knowledge and back into explicit knowledge so that a knowledge repertoire should be developed that may facilitate a smooth transfer of knowledge from university to industry.

## Limitations and future research

There are also several limitations of this study that should be considered while interpreting the results. First, as most of the prior studies, this study employed a lagged research design in order to minimize issues related to the subjective and method biases. There should be longitudinal research in the future to examine the hypothesized model. Further, our study is based on the premise of entrepreneurial knowledge. As new knowledge will be accumulated, there are chances that students' AC with respect to GEB will change over time. In this regard, we suggest future studies to adopt a mixed methodology or a quasi-experimental research design to gauge

the phenomenon. Third, our study employed the AMO theory to build the relation among variables. However, external factors may influence opportunities that facilitate students to effectively commence a green enterprise, e.g., family support, financial support, prior industrial experience, etc. Therefore, future research studies may extent the boundary conditions of GEB. Fourth, the relationship between IS and GKT and GEB may also be intervened by some other contingent factors, which may also be studied in the future research. Last but not the least, although there is a growing consensus that supports university students pursuing entrepreneurial careers. Future research may include people from different backgrounds, age groups, and cultural background to provide a finer-grain understanding of the phenomenon.

## Conclusion

In conclusion, this study offers vital insights into the dynamics that influence GEB among university students. Our findings highlight the significant roles of IS and GKT as key antecedents in enhancing GEB. Moreover, the study uncovers the crucial mediating role of AC in this relationship, demonstrating how the internalization of green knowledge and support enhances students' engagement in green entrepreneurship. Additionally, the moderating effect of ER further elucidates the nuanced ways in which personal commitment to environmental issues interacts with institutional and knowledge factors to influence GEB. These results not only contribute to the academic discourse by expanding the understanding of factors driving GEB but also offer practical implications for educational institutions and policymakers in fostering green entrepreneurial initiatives among youth. By focusing on the enhancement of IS and GKT, alongside nurturing AC and acknowledging the influence of ER, strategies can be developed to effectively cultivate a new generation of environmentally responsible entrepreneurs.

## Author Contributions

**Conceptualization:** Yejun Yang.

**Formal analysis:** Yejun Yang.

**Methodology:** Yejun Yang.

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
