## [Decision Letter · Decision Letter 0]

15 Nov 2023

PONE-D-23-26917Impact of institutional support and green knowledge transfer on university students’ absorptive capacity and green entrepreneurial behavior: the moderating role of environmental responsibilityPLOS ONE

Dear Dr. Yang,

Thank you for submitting your manuscript to PLOS ONE. After careful consideration, we feel that it has merit but does not fully meet PLOS ONE’s publication criteria as it currently stands. Therefore, we invite you to submit a revised version of the manuscript that addresses the points raised during the review process.

Your paper has been reviewed by experts in their fields and they have shown interest in your paper, however, they have also highlighted some minor changes. Therefore, I request you to address the reviewers' comments and submit the revised version after necessary modifications. 

We look forward to receiving your revised manuscript.

Kind regards,

Arslan Ayub

Academic Editor

PLOS ONE

Reviewers' comments:

Reviewer's Responses to Questions

**Comments to the Author**

1. Is the manuscript technically sound, and do the data support the conclusions?

Reviewer #1: Yes

Reviewer #2: Yes

2. Has the statistical analysis been performed appropriately and rigorously? 

Reviewer #1: Yes

Reviewer #2: Yes

3. Have the authors made all data underlying the findings in their manuscript fully available?

Reviewer #1: Yes

Reviewer #2: Yes

4. Is the manuscript presented in an intelligible fashion and written in standard English?

Reviewer #1: Yes

Reviewer #2: Yes

5. Review Comments to the Author

Reviewer #1: Impact of institutional support and green knowledge transfer on university students’ absorptive capacity and green entrepreneurial behavior: the moderating role of environmental responsibility

Thank you for granting me the opportunity for review. Very interesting and new insight in the way of scientific research on the role of environmental responsibility especially in the emerging nations. Sufficient literature is applied, and the research gap is articulated in a good way. Methodological tools have been adequately applied. The theme of the paper is very interesting and has a lot of practical implications. Overall, the paper is well-written and can be considered for publication. However, the following are some minor suggestions to improve the paper.

• Abstract should be improved in the following way. First, you follow the aims/Objective. Second, you should follow the methodology. Third, you should follow the findings and conclusion drawn from their findings. Fourth implications.

• Introduction should be developed scientifically. First, you discuss the problems and their significance. The clear objectives and questions should be mentioned. The contribution of the study should be mentioned clearly,

• There are a lot of grammatical mistakes, authors need to improve.

• Justified your findings with previous literature that discussed in the literature section.

Best of Luck

Reviewer #2: First of all, I appreciate the author contribution towards novelty. The paper is well written but I have few suggestions for authors as per my expertise.

1) abbreviation used in abstract portion that is not meaningful, you can explain them first and don't use any abbreviation in abstract portion specifically.

2) The literature review segment till analysis is perfectly written but conclusion portion is missing.

3) Specify more literature about green entrepreneurship.

6. PLOS authors have the option to publish the peer review history of their article (what does this mean?). If published, this will include your full peer review and any attached files.

Reviewer #1: No

Reviewer #2: **Yes: **Dr. Zeeshan Rasool

---

## [Author Response · Author response to Decision Letter 0]

23 Feb 2024

Reviewer #1: Impact of institutional support and green knowledge transfer on university students’ absorptive capacity and green entrepreneurial behavior: the moderating role of environmental responsibility

Thank you for granting me the opportunity for review. Very interesting and new insight in the way of scientific research on the role of environmental responsibility especially in the emerging nations. Sufficient literature is applied, and the research gap is articulated in a good way. Methodological tools have been adequately applied. The theme of the paper is very interesting and has a lot of practical implications. Overall, the paper is well-written and can be considered for publication. However, the following are some minor suggestions to improve the paper.

• Abstract should be improved in the following way. First, you follow the aims/Objective. Second, you should follow the methodology. Third, you should follow the findings and conclusion drawn from their findings. Fourth implications.

• Introduction should be developed scientifically. First, you discuss the problems and their significance. The clear objectives and questions should be mentioned. The contribution of the study should be mentioned clearly,

• There are a lot of grammatical mistakes, authors need to improve.

• Justified your findings with previous literature that discussed in the literature section.

Best of Luck

Response 

Thank you for your constructive feedback and encouraging comments on our manuscript. I have made the following revisions in accordance with your suggestions:

1. I have restructured the abstract to sequentially include the aims/objectives, methodology, findings, and implications.

2. The introduction has been scientifically developed to more clearly articulate the problem, its significance, and the study's specific objectives and contributions.

3. I have thoroughly reviewed the manuscript and corrected grammatical errors to improve the overall quality of the writing.

4. The findings have been more explicitly justified and linked with the existing literature discussed in the literature review section.

Reviewer #2: First of all, I appreciate the author contribution towards novelty. The paper is well written but I have few suggestions for authors as per my expertise.

1) abbreviation used in abstract portion that is not meaningful, you can explain them first and don't use any abbreviation in abstract portion specifically.

2) The literature review segment till analysis is perfectly written but conclusion portion is missing.

3) Specify more literature about green entrepreneurship.

Response 

I greatly appreciate your positive remarks and valuable suggestions. In line with your recommendations:

1. I have revised the abstract to explain the key terms initially and avoided the use of abbreviations, thereby making it more accessible and understandable.

2. A conclusion section has been added to effectively summarize the key findings and their implications.

3. More literature on green entrepreneurship has been incorporated to enrich the theoretical underpinnings of the study.

I believe these revisions have significantly enhanced the quality of our manuscript and thank you for your insightful guidance in this process.

---

## [Decision Letter · Decision Letter 1]

7 May 2024

Impact of institutional support and green knowledge transfer on university students’ absorptive capacity and green entrepreneurial behavior: the moderating role of environmental responsibility

PONE-D-23-26917R1

Dear Dr. Yang,

We’re pleased to inform you that your manuscript has been judged scientifically suitable for publication and will be formally accepted for publication once it meets all outstanding technical requirements.

Kind regards,

Arslan Ayub

Academic Editor

PLOS ONE

Additional Editor Comments (optional):

Reviewers' comments:

Reviewer's Responses to Questions

**Comments to the Author**

1. If the authors have adequately addressed your comments raised in a previous round of review and you feel that this manuscript is now acceptable for publication, you may indicate that here to bypass the “Comments to the Author” section, enter your conflict of interest statement in the “Confidential to Editor” section, and submit your "Accept" recommendation.

Reviewer #1: All comments have been addressed

Reviewer #2: All comments have been addressed

2. Is the manuscript technically sound, and do the data support the conclusions?

Reviewer #1: Yes

Reviewer #2: Yes

3. Has the statistical analysis been performed appropriately and rigorously? 

Reviewer #1: Yes

Reviewer #2: Yes

4. Have the authors made all data underlying the findings in their manuscript fully available?

Reviewer #1: Yes

Reviewer #2: Yes

5. Is the manuscript presented in an intelligible fashion and written in standard English?

Reviewer #1: Yes

Reviewer #2: (No Response)

6. Review Comments to the Author

Reviewer #1: All points addressed by the authors and i recommend the to accept the manuscript. The quality of paper is good and the journal should include for the publications.

Reviewer #2: I appreciate the researcher findings and interest for specific research objectives. The author addressed all my concern points and queries.

7. PLOS authors have the option to publish the peer review history of their article (what does this mean?). If published, this will include your full peer review and any attached files.

Reviewer #1: **Yes: **Muhammad Waris

Reviewer #2: No

---

## [Editor Report · Acceptance letter]

9 May 2024

PONE-D-23-26917R1 

PLOS ONE

Dear Dr. Yang, 

I'm pleased to inform you that your manuscript has been deemed suitable for publication in PLOS ONE. Congratulations! Your manuscript is now being handed over to our production team.

Kind regards, 

on behalf of

Dr. Arslan Ayub 

Academic Editor

PLOS ONE